# Raising the Child—Do Screen Media Help or Hinder? The Quality over Quantity Hypothesis

**DOI:** 10.3390/ijerph19169880

**Published:** 2022-08-11

**Authors:** Diana Puzio, Iwona Makowska, Krystyna Rymarczyk

**Affiliations:** 1Department of Child and Adolescent Psychiatry, Medical University of Lodz, 90-419 Lodz, Poland; 2Department of Biological Psychology, University of Social Sciences and Humanities in Warsaw, 03-815 Warszawa, Poland

**Keywords:** screen time, development, children, digital health

## Abstract

Screen media are ubiquitous in human life across all age, cultural and socioeconomic groups. The ceaseless and dynamic growth of technological possibilities has given rise to questions regarding their effect on the well-being of children. Research in this area largely consists of cross-sectional studies; experimental and randomized studies are rare, which makes drawing causative conclusions difficult. However, the prevailing approach towards the use of screen media by children has focused on time limitations. The emerging evidence supports a more nuanced perspective. It appears that the older the child, the more important how the screen media are used becomes. Concentrating on the quality of the screen, time has become increasingly relevant in the recent COVID-19 pandemic, which necessitated a transfer of educational and social functioning from real-life to the digital world. With this review, we aimed at gathering current knowledge on the correlations of different screen media use and development outcomes, as well as providing an overview of potential benefits that new technologies may provide to the pediatric population. To summarize, if one cannot evade screen time in children, how can we use it for children’s maximum advantage?

## 1. Introduction

The debate over whether exposure to screen media has any impact on child development and, if so, the nature of this impact, is unresolved despite decades of research [1].

During the past quarter of a century, new technologies have distinctly influenced the way children and adolescents communicate with others, familiarize with their environment and learn about the world [2]. While screen media constitute an embedded element of the milieu in which children and adolescents thrive, a question arises as to how new technologies permeate the process of development. Some wonder whether new technologies, at least partially, account for observed phenomena in the area of children’s and adolescents’ development, as well as possible mental and physical health problems [3]. 

Correlational studies indicate that excessive screen time in children (we understand excessive screen time as media consumption exceeding AAP recommendations [4]) is associated with poorer cognitive, linguistic and socio-emotional skills [5]. A recent review points to similar negative aspects of technology use concerning adolescents’ cognitive and psychosocial development [6]. 

This debate is urgent, especially considering the COVID-19 pandemic, which resulted in increased screen time in all age populations, especially in children [7], and more parental mental health problems which were positively correlated with excessive screen time and the use of mature-rated programming for children [8]. Moreover, media use by parents may have additional detrimental effects on children’s development by interfering with parent–child interactions. A recent experimental study revealed decreased neural synchrony between mothers and eight girls aged 24–42 months during dialogic reading interrupted by smartphone usage in comparison to dialogic reading without such breaks [9]. The paper focuses on human early development; however, this process may influence or precipitate the origins of mental and physical health problems. Indeed, scientists point to the increasing, during COVID-19 era, negative link (or influence) of screen media use to several health problems in the pediatric population, e.g., obesity [10], mental health problems [11], diminished physical activity [12], unhealthy eating [13], headache prevalence [14] and sleep disorders [14]. However, elaboration on this topic is beyond the scope of this review. 

The debate on whether the magnitude of exposure to screen media has any associations with developmental disturbances (which may pose a risk or trigger following mental and physical health problems) continues. The aim of this paper is to provide a review of relevant studies shaping the dispute. Henceforward, we discuss cognitive, motor and emotional aspects of development in the digital era, as well as an overview of selected digital interventions designed to support mental and physical health. 

The term “child” is used to refer to the population in the prepubertal developmental age, beyond this age the term “adolescent” is used. In the description of particular studies, we aimed at giving concrete information on age groups or developmental stages of examined participants. We would like to add a disclaimer that for the majority of this paper, the child or the adolescent represents a member of middle-class European or Northern American society because these are the geographical areas which provide most of the body of research. Studies incorporating various cultural and geographical origins, preferably accounting for these confounders, are warranted.

## 2. Cognitive Development—Language and Executive Functioning

Beyond the age of 2, studies examining the developmental impact of screen media yield conflicting results, demonstrating positive, negative or no associations, depending on the age of a child, type of device, program and contextual factors. Various research within this area concentrates particularly on the associations between screen device use and linguistic skills evolution in infants and toddlers. For example, one study’s results suggested that toddlers were more likely to learn new words from interactive touchscreen devices than from television programming [15]. These results are in line with research demonstrating that children learn language through responsive and interactive exchanges rather than by passive viewing [16,17]. In one experimental study toddlers aging from 15 to 24 months old were taught new words via child–adult interactions versus the *Teletubbies* TV program. Results showed that, until 22 months of age, children were not able to learn new words from TV programs. Authors concluded that in the prelinguistic and newly verbal state of development, children need human interaction to acquire vocabulary [18]. Accordingly, it is claimed that there is a negative association between television viewing in this population and language and executive functioning development [19,20,21]. In one well-designed, cross-sectional study of children aged from 12 to 35 months, excessive TV viewing was significantly linked to delays in cognitive and language development skills measured with Bayley Scales of Infant Development-second edition [22]. In fact, even background television has been associated with disruption of sustained toy play in 12 and 24-month-old children. Additionally, it was linked to a reduction in the quantity and quality of parental engagement in play, especially parental verbal expressions towards their children [23,24]. A recent meta-analysis including 42 studies concluded that overall screen time and background television was associated with poorer language outcomes, while educational programming, co-viewing and later age of screen use onset was linked to better language skills in children [25]. On the other hand, a well-designed observational study examining the associations between viewing specific television series and language development trajectories in infants from the age of 6 to 30 months old reported more nuanced results. Watching programs that embedded strategies promoting vocabulary acquisition and expressive language—obtaining viewers’ attention, speaking directly to the viewer, providing opportunities for the viewers to respond and labeling particular objects—were linked with more expressive speech and vocabulary than other programs. It seems that programs such as *Arthur* and *Clifford* act by eliciting the direct participation of the viewer and come closer to the holy grail of language development: interaction. The beneficial effects of *Sesame Street* viewing were not observed in this study, which is contradictory to research encompassing preschool children. This might be associated with the obvious age difference but also, as authors declare, the data were collected prior to the changes in the structure and curriculum of the program [26].

Some research underscores the importance of the bidirectional transfer deficit of acquired knowledge or skills from two-dimensional material to 3D world [27]. Results from various studies suggest that the transfer deficit decreases with age yet persists in 2 and 3-year-old children [27,28]. It seems that it can be ameliorated by socially contingent interactions, i.e., live interactions and video chats [29], through verbal and non-verbal cues [30]. 

Current guidelines suggest no screen time for infants below 24 months old (except video chatting) [4]. This stands in contrast with parental reports of media consumption of their children below the age of 2 years old. Educational interventions explaining current guidelines and the limited profits of screen time in infants may enhance better adherence to recommendations [31]. Additionally, the harm reduction approach has been suggested to implement in practical milieu to overcome the impasse of prevailing recommendations that will most likely not be followed in the era of constant media presence, lockdowns and separation from human interactions [32]. 

On the other hand, some scholars argue that condemnation of any screen media use by infants may simplify recommendations at the expense of conveying more detailed and comprehensive information. There is a growing body of evidence that the content and time of screen media is not all that matters in terms of cognitive outcome. For example, an active participation of an adult emphasizing the content of screen media improved the retention and later transfer of information in infants of 12 and 18 months old [33]. Moreover, scholars found that 2-year-olds who were exposed to repetitive screen stimuli enhanced performance in the context of imitation and vocabulary acquisition [34]. Other factors that should be taken into consideration are the quality and quantity of parent–child interactions independent of screen media use. Parental television viewing, internet use and background television are prevalent and most likely displace the interpersonal interactions with children [35]. The aforementioned studies prompt the assumption that future research should examine the various environmental factors influencing language development simultaneously in order to account for possible confounding factors. As long as screen media use is examined as an individual, separated factor influencing children’s development, the conclusions can be hardly transferred from controlled, experimental settings to the real-life environment.

As far as older children are concerned, in the study of almost one hundred children aged from 36 to 60 months old, screen time had no link with expressive vocabulary. However, excessive screen time did correlate with poorer working memory [36]. Some studies, mostly experimental and short-term in design, claim that playing video games may have the potential to improve executive functioning in children. One study found an enhancement of working memory in preschool children after 5-week computerized training in comparison to children playing regular computer games. No significant changes were observed in inhibitory skills [37]. However, another study revealed the improvement of inhibition and attention in elementary school children who were also exposed to computerized training. Their executive functions were improved after 10 weeks of one to three sessions per week of playing for only 15 min. These results were reflected in better math and literacy grades at the end of the year [38]. A different study investigated whether playing video games and non-computerized games for 1 h twice per week for 8 weeks was associated with the improvement of different cognitive skills. It revealed that children exposed to games rewarding quickness improved in terms of processing speed, while those exposed to games rewarding reasoning improved on tasks exploiting cognitive reasoning. Unfortunately, due to the mixed methodology, one cannot distinguish the effect of digital and non-digital tasking [39]. Moreover, scientists point to the greater importance of contents than duration of screen exposure. It is underscored that high quality, age-appropriate programming may have a beneficial effect on children’s cognition, whereas developmentally inadequate video games, shows or movies pose a risk to mental and health problems [40]. Indeed, the authors of the meta-analysis comprising 24 studies of various socioeconomical groups examining the effect of *Sesame Street* watching on preschool children concluded that it was positively associated with better literacy and numeracy abilities and improved knowledge about the world, which could mirror a greater readiness for school. The results were significant regardless of socioeconomical status, which seems important since underprivileged societies may have less opportunities to stimulate the cognitive development of the young population [41]. One of the few longitudinal studies in the field checked the screen time (both watching and playing) in the group of adolescents at the age of 11, 15 and 18 years old and measured working memory at the age of 22. Surprisingly, more screen time at the age of 11 and 15 was positively correlated with working memory, but only in boys and the effect was mediated by IQ [42]. In line with these results, a longitudinal cohort study encompassing 9 to 10-year-old children showed that intelligence was negatively correlated with watching and gaming time at the baseline. However, after two years of observation, gaming had a beneficial impact on intelligence both in boys and girls. Surprisingly, the same effect was observed for watching activities; however, it disappeared after accounting for parental education. 

On the other hand, other studies show the detrimental effects of excessive video game playing and watching videos on executive functioning, underlining the benefits of physical activity in school-aged children [43,44]. Indeed, even background TV exposure while sleeping was shown to correlate with decreased executive functioning in preschool children in a survey-based cohort study [45]. One recent study applying a specific tool to measure visual and narrative demands of TV shows revealed that poorer executive functioning in children is linked to programming of greater stimuli content, more challenging in terms of comprehension and with a quicker pace [46]. A study on adolescents revealed that the lack of sleep may play a mediating role in the relationship between video game playing and executive functioning deficits, particularly attention [47]. In fact, playing video games before bedtime was associated with poorer academic attainment in boys [48]. These results were confirmed by a recent meta-analysis. It revealed that television viewing and video game playing was linked to worsened academic performance, while overall screen time was not [49]. However, a longitudinal cohort study of over 1000 children showed that time spent watching TV was negatively associated with attention skills in adolescence. Data were collected every two years from the age of 5 to 15 years old and the results were significant regardless of gender, attention problems in early childhood, cognitive ability at the age of 5 years old, and childhood socioeconomic status, as well as adolescent television viewing [50].

Taken together, it seems that playing video games in an experimental environment enhances executive functioning, measured both in the short-term and long-term perspective. In fact, some suggest that the Flynn effect (gradual increase in intelligence scores in the last few decades) was time-correlated with the introduction of video games and they may constitute a technological explanation for this phenomenon [51]. Although Scandinavian countries, which are technologically advanced and have high rates of gaming, began to report an inverse of the Flynn effect [52]. However, studies show that children and adolescents devote much more screen time to video playing in real life than in studies [7]. This may be a reason for links between poorer academical performances and videogame playing reported in the aforementioned meta-analysis. Further longitudinal studies are warranted to better understand the relationships between different video games, playing time and cognitive outcomes in children and adolescents [53]. 

In conclusion, future approaches could compare curriculum-based television programs such as *Sesame Street* or slow-paced narrative series with other entertainment programs to further examine the hypothesis of the leverage of quality over quantity of screen use. Strong and causative evidence could be elicited from experimental studies manipulating the exposure to particular programs. The results of such comparative studies would perhaps be helpful to create a standard of good quality programs for children in different age groups. In addition, a lot of research concentrates on results measured with psychological tests; it would be interesting to examine in observational studies how screen device use impacts real-life performances depending on the environmental and cultural challenges, e.g., achieving a higher education, ability to run a family business or obtaining satisfactory employment. Such knowledge might be helpful for parents and caregivers in particular environmental circumstances. 

## 3. Motor Development

Infancy and childhood are critical periods for developing fundamental motor skills. Greater competency in gross motor skills is associated with higher levels of physical activity and better perceived physical competence in adolescence [54,55], as well as maintaining adequate body mass [56]. Fine motor skills competence, on the other hand, was linked to higher performance intelligence [57] and better academic achievements [58]. 

Some scholars suggest that children’s motor skills have been recently deteriorating when compared to normative ranges [59], while sedentary behaviors exceed recommended limits in children worldwide [60]. Growing screen time in the pediatric population has led scientists to investigate whether there are any links between two phenomena. One study revealed that in 18-month-old and 24-month-old children, the emerging gross motor skills are negatively correlated with the time spent on sedentary activities [61]. Conversely, a cross-sectional study with over one hundred children who were 3 to 5 years old revealed that smartphone use was positively associated with fine motor skills in 3-year-old children; no associations were established for older children [62]. 

In the aforementioned cross-sectional study from Taiwan, researchers compared the motor skills of children exposed to excessive screen time with those who met the then recommended limits (no screen time below the age of 2 and a maximum of 2 h of screens above that age). A Peabody Developmental Motor Scales-second edition (PDMS-2) test was used to assess fine and gross motor skills. The study showed that there were two predictors for motor development delays, namely excessive screen time and double-income family. Although the relationship was not significant, children watching TV excessively were almost two times more likely to have motor development delays [22]. Moreover, one longitudinal study found that greater engagement in screen device use at the age of 4 and 5 was associated with poorer competence in fundamental motor skills at the age of 7 [63]. A study of children from 4 to 7 years old demonstrated that total screen time was significantly, negatively correlated with visual-motor integration, in-hand manipulation, sensory processing and bilateral coordination skills. Importantly, playing with toys and object substitution in play were potential moderators of these associations, loosening the links [64]. Similarly, an experimental study assessed the fine motor skills of preschool children before and after 24 weeks of fine motor skills 24-week training program, either on the tablet (study group, *n* = 40) or with manual tasks (control group, *n* = 40). At the baseline, there were no significant differences in fine motor skills between the two groups; however, after the training was completed, the control group performed better, especially in terms of precision, integration and manual dexterity, compared to the study group [65]. It has been suggested though that this might have been the effect of the tool used to measure fine motor skills, which required skills exercised during the training of the control group. In line with those findings, an observational study of pre-school children demonstrated that those with higher computer use had worse outcomes of locomotor skills and overall fundamental motor skills [66]. Results from a recent prospective observational trial suggest that screen time was positively associated with the worsening of fine motor skills, especially those responsible for writing and drawing. These findings were more pronounced in boys, although screen time did not differ within genders [67]. Additionally, results from this study add to the existing evidence of the link between physical activity and motor competence that are crucial for children’s health [67]. 

However, a cross-sectional study of children of 24 to 42 months old showed results contrary to previous studies. Children who used tablets on a daily basis (from 10 to 120 min per day) had better results in Bayley-III Fine Motor Scaled scores than children who had no experience in tablet use. Authors noted that children were mostly engaging in passive activities such as watching videos; however, playing games was the next activity in line. Importantly, almost 77% used the device always under parent supervision [68]. In another study of preschool children, adjusted time spent on interactive video games was associated with more competence in object control skills but not locomotory skills [69]. In fact, preliminary evidence supports the use of *iPad* applications in order to improve fine motor skills in preschool and elementary school children, including those with special need. Such interventions seem not only to be beneficial but also to evoke in participants more motivation and engagement than traditional occupational therapy [70,71].

To conclude, studies to date examining associations between screen device use and motor development give conflicting results. Nonetheless, the observational design of these studies does not allow to distinguish cause from effect. Further methodological limitations include different assessment methods and short-term design. Further research is warranted to determine the influence of different device use on physical competency, as well as mechanisms of possible impacts. 

## 4. Emotional and Social Development

One of the major developmental tasks in childhood is learning how to regulate emotions, control behaviors and socialize [72]. Mastering self-regulatory skills was associated with educational success, resilience and wealth [73]. By contrast, failing in obtaining competence in a self-regulatory area may pose a risk for future adverse health outcomes [74] and predicts criminal history [73]. It has been hypothesized that screen media exposure in early childhood may influence the process of acquiring self-regulatory skills [75]. Scholars suggest that early onset of device use may decrease chances for developing internal regulatory mechanisms when a screen media is offered as a distraction tool [76]. Moreover, using screen devices displaces responsive parental interventions and parent–child interactions, which have been associated with better self-regulatory characteristics [77].

Indeed, current evidence suggest that children often presenting fussy behaviors, such as having trouble with sleep and feeding, watched more television than children without such problems [78]. One longitudinal cohort study of infants aged 9 months and toddlers at the age of 2 years old not only confirmed these associations but also indicated that they were more pronounced in children of lower socioeconomical status [79]. It appears that children in families of lower income are more likely exposed to screen media as a relatively safe and easily available method to calm the child down. In line with these findings, a recent study demonstrated that socioeconomical status accounted for almost half of the self-regulation and media exposure links [80]. Research indicates that another moderator of discussed associations may be parental strain that leads parents to use screen media when they feel overwhelmed and not capable of coping with children’s difficulties themselves [80,81,82]. However, scholars report that using media devices as a tool to regulate children’s emotionality and behavior may in fact lead to increases in negative emotionality, creating a vicious circle of inadequate parental interventions and augmenting self-regulatory problems. Indeed, emerging evidence implies that the relationship between screen exposure and regulatory difficulties is bidirectional. A longitudinal study of 2, 4 and 6-year-olds revealed that smaller media consumption in toddlers was associated with better self-regulatory competence at the age of 4, while lower self-regulatory skills at this age predicted more screen time and television viewing at the age of 6 [83].

The discussed results present several drawbacks. Firstly, they come largely from cross-sectional studies that do not directly address the question on causality. Furthermore, research to date investigated mainly television viewing, ignoring the growing adoption of other, mainly hand-held, devices in children. Additionally, studies did not distinguish different programming (e.g., educational vs. adult-oriented) and whether parents actively participated in screen time serving guidance. It seems important to examine these issues as they have been found to moderate developmental outcomes of screen exposure in children [84,85]. 

As far as social development is concerned, the body of research is limited. Scholars seem to agree that fewer daily interactions, both with peers and caregivers, are related with delays in social development [86]. It has been implied that screen time displaces social activities and may inhibit development in this area. A recent study comprising children between 1, 5-year-old and 3-year-old reported that there is an association between the use of touch screen devices and social withdrawal and aggressive behavior [87]. Moreover, television exposure and less interactive play with a parent during infancy was positively correlated with autistic-like symptoms at the age of 2 [88,89]. Associations of poorer social development and screen time have been also obtained in the group of preschool children. A survey-based study of almost 600 children aged 2 to 5-year-old found an adverse association between the daily amount of television or video viewing and children’s social skills. According to previous reports, outdoor play had positive associations with social competency outcomes [90]. On the other hand, video-chatting enables live face to face interactions with relatives that may be otherwise impossible due to, e.g., epidemiological reasons. In such instances, allowed by current recommendations, even for children below 18 months of age, interactions via screen devices may help to develop and maintain close relationships [91]. It seems plausible that children in the prelinguistic stage might react more readily to the face on the screen and changing face expressions rather than verbal communication. It would enrich our understanding of how children experience live video-chatting if it was compared to, for example, traditional phone calls. The results coming from one study encompassing young women suggests the superiority of video-chatting over other means of distant communication [92]. 

Imitative behaviors, such as the ability to learn new skills by copying others, is another aspect of social development that has undergone vivid discussion in terms of screen media use. Historically, the prevailing notion was that children learn less from screen agents [93]. However, newer studies suggest that the potential to elicit imitative responses of screen-based agents may be similar to that of in person interactions, since these digital agents become more and more humanoid and appealing [94]. In fact, one study of preschool children found that they were keen to employ ineffective behaviors suggested via a demonstration video while remaining immune to such suggestions when demonstrated in face to face interactions. This may indicate that children transfer their previous experience with digital media as trustworthy and interpret information from digital devices as credible [95]. Fortunately, studies show that children display the ability to evaluate incoming information both from live and digital sources, based on previous inaccuracy. This emphasizes that children need a competent adult to guide their use of screen media [96].

As far as older children are concerned, an increasing number of scientists point out that concentration on screen time may not be a reliable method to assess relations between emotional and social functioning and screen media. On the contrary, they encourage focusing on how the media are used, a more nuanced approach in comparison with a singular measurement of screen time [97,98,99]. For instance, emerging evidence supports the idea that it is the nature of social networking sites activities that adolescents engage in that influences their well-being. Study results point to associations between less anxiety and positive online interactions that enhance social support and facilitate socializing in person. Conversely, negative interactions and social comparison online, as well as the reduction in face-to-face experiences, appears to be connected with more internalizing symptoms [100,101,102]. It is also being underscored increasingly often that children are affected differently by screen media depending on their individual characteristics (e.g., adverse experiences, health problems or mental health problems, socioeconomic status), which has not been reflected in robust cohort-studies concerning the rise in screen time fixation. For example, a history of prior victimization was predictive of experiencing cyberbullying among adolescents [103] and mental health problems seem to mitigate seeking out negative content online and passive viewing [104]. Moreover, children who encountered cyberviolence may by prone to become a perpetrator themselves [105]. In addition, children from disadvantaged environments appear to be more prone to negative experiences online, while those coming from well-resourced families display adaptive behaviors online and report positive experiences [106,107].

The evidence coming from the majority of studies is poor in effect and correlational, giving no possibilities for discriminating cause from effect. Studies use mainly self-report data which are prone to social-desirability bias and inaccuracy, both in qualitative and quantitative evaluations. Additionally, many studies fail to account for possible confounding factors such as socioeconomical status or preexisting mental health problems (both diagnosed and undiagnosed) in children and adolescents, which may easily be responsible for observed results. Future studies would enable causative conclusions by adding randomized exposure to media, preferably with the blinding of participants. The use of in-built applications to objectively measure not only time but, primarily, how screen media are used (e.g., Human Screenome Project) [108] could provide researchers with data that reflects the complexity of screen media use associations. The limitation to that would be privacy concerns. Lastly, participants included in future research should undergo careful and in-depth examination prior to the study to account for substantial heterogeneity in social and emotional functioning, which may not be reported in a basic survey of socioeconomical status or be withheld if the survey is conducted in public circumstances. For example, future research should incorporate an assessment of what is the place of new technologies in the lives of children and adolescents of racial and LGBTQ+ minorities, as well as neurodevelopmental or psychological/psychiatric disorders, especially determining specific risks and potential benefits for these populations. To conclude, it seems that credible research in the field of digital device use and its associations with the development of children and adolescents requires a multidisciplinary team of specialists from the field of technology science, health professionals and psychologists.

## 5. Digital Health Interventions

The interest of possible prophylactic and therapeutic use of digital devices and programming in the pediatric population is correlated with the growing ubiquity and economical availability of new technologies. Different measures and patterns are exploited to design health-promoting programming for children and adolescents. Henceforth, we present an overview of proposed digital health interventions, divided in terms of specific programming.

For example, a recent systematic review incorporating several randomized controlled trials showed that the adolescent use of mobile phone applications (apps) for several chronic diseases may improve the adherence to treatment. Investigated apps remind the patient of medication time through automatically generated text message, ameliorating treatment outcomes of acne, asthma, diabetes mellitus, depression, human immunodeficiency virus infection, liver transplant, sickle cell disease and systematic lupus erythematosus [109]. Over three hundred apps have been retrieved to enhance attention deficit and hyperactivity disorder. However, a systematic review found only one study examining these apps that met the inclusion criteria, thus no conclusions as to their efficacy and safety were established [110].

In fact, the body of research on health apps lacks experimental trials of specific apps but is mostly based on general descriptions. Accordingly, a recent paper on apps meant to treat speech disorders in children presented several applications of good-quality which have the potential to enhance therapy, particularly when in-person interventions are limited [111]. Another review of apps advertised as beneficial in a changing diet, physical activity and sedentary behaviors patterns in children and adolescents concluded they may be of value in supporting patients through several techniques: instructions, general encouragement, contingent rewards and feedback on performance [112]. Researchers from Iran concluded that apps may provide the population of pediatric oncological patients and their parents with reliable information on symptoms assessments, social support and include calendar reminders [113]. 

Although the majority of research focuses on potential for harm of video game playing [114], “serious games” or “games for health” have been developed and trialed to actually improve children and adolescents’ health through playful activity and entertainment. *Re-Mission*, *Personal Investigator*, *Treasure Hunt* and *Play Attention* are some of the examples of games dedicated to enhancing oncological and psychotherapeutic treatment. Medical and public attention stemming from this area of research has been mirrored in the emergence of a new journal devoted solely to games’ use for therapeutic reasons: *Games for Health Journal* [115]. Indeed, a systematic review of the literature regarding several chronic diseases in the pediatric population, i.e., cerebral palsy, asthma, diabetes, developmental coordination disorders, and vision disorders, found mixed results with no potentially beneficial outcomes of therapy when serious games were utilized [116]. Interventions based on serious games playing also demonstrated effectiveness in enhancing knowledge and self-management in pediatric patients with diabetes, asthma and cancer [117]. Furthermore, a recent meta-analysis of seven randomized controlled trials concluded that playing virtual reality games can improve gross motor skills in children with cerebral palsy [118]. Two systematic reviews evaluating serious games’ impact on psychotherapy results found that games were mainly based on cognitive behavioral techniques and were used either as a separate intervention or alongside traditional psychotherapy. It has been suggested that at times, playing video games may yield similar results as a traditional psychotherapy and may be more acceptable than traditional, established treatments to a subset of clients. The authors of these meta-analyses implied that games for health proved some effectiveness, but the results were not conclusive, and more research is needed [119,120]. Additionally, while excessive screen time and digital device use have been proven to cause obesity in the pediatric population, there is emerging evidence of contrary effects of games requiring gross motor activity: exergames. The growing body of research indicates that playing exergames by overweight and obese pediatric patients may increase their motivation for physical activity; however, their impact on actual body weight is yet to be established [121,122].

Another emerging area of research concerns the utilization of social networking sites in order to prevent suicidal behaviors through monitoring social media for certain terms or improving mental health in adolescents with depression or psychosis by online social therapies. However, the broad implementation of these programs is inhibited by the paucity of research and constraints about privacy policy [123,124].

Despite promising results, scholars underscore that research in the area of serious games lacks methodological rigor that would prompt final recommendations. Limitations include the predominance of quasi-experimental, short-term designs, survey-based data collection, small samples and ignoring measurement of possible negative effects of game playing. 

It seems that examining individual applications with a scientific rigor implemented in medical trials of new interventions, i.e., randomization, blinding, accounting for confounding (group heterogeneity), is warranted to recommend these programs as true interventions promoting or serving health. Collective assessments of DHI might be futile, similarly to general estimations of the effectiveness of a group of certain drugs. Finally, we argue that cooperation with specialists of digital science such as artificial intelligence and programming is necessary throughout the study to ensure safety measures and the correct implementation of the product. It would be best if these specialists were independent of the product owners and manufacturers.

## 6. Conclusions

As screen devices permeate almost every aspect of life and accompany human life from the very beginning, questions concerning functional duration of exposure as well as the mode of use by children have increased. It seems that, as in other areas of familiarizing with the world, parents and other caregivers should be the informed guides and moderators of interactions with screen devices.It seems that there is not enough evidence to suggest that learning from videos is a better option for infants and toddlers than engaging them directly. However, certain features of age-appropriate programs and their use such as parental scaffolding, embedding social cues and repetition may ameliorate acquiring language abilities. Screen time in the youngest population might be also considered in terms of harm reduction interventions targeting parents’ education. Simply forbidding the use of screens may evoke parental guilt and further the emergence of vicious circle mechanisms. Supporting parental engagement in play and other interactions with children is crucial. Informing care givers on the content that has minimal or no harm potential for the youngest children may prevent the usage of age-inappropriate and deleterious programming.Playing computer games may improve executive functions in children, especially games that were designed for that purpose. However, studies show that yielding positive results in this field can be achieved already by low to moderate screen exposure.The data on the links between motor development and screen media use is inconclusive. However, there is emerging evidence of a potential to facilitate the development of fine motor skills in children by using touchscreens. Although it is of equal importance that screens cannot substitute the real-life object manipulations and interactions. They may serve as an additional tool exploited under certain conditions: parental monitoring and moderation, time limitations, exposure to active programming (i.e., games, educational applications) and preventing any inappropriate content.Excessive screen media use is correlated with delays in gross motor skill development in younger children and less physical activity in older children and adolescents. This, in turn, poses a risk to obesity, cardiovascular disorders and other diseases of civilization. However, certain applications or games may encourage children and adolescents to engage in more physical activity. Digital media may serve as a provider of positive role models or the opportunity to exercise (exergames)Caregivers should be vigilant not to offer screen devises as a reward, soother or a replacement for surveillance. Such an attitude was associated with poor emotional development and problems with self-regulation.Adolescents seem to reflect acquired social skills and off-line functioning in their online activities. This warrants targeting vulnerable individuals to prevent further isolation or victimization. Time-focused approaches in research may be futile in this age group, especially when experiences with lockdowns and school closures during the COVID-19 pandemic are taken into consideration. We would suggest future research to include interviews or examinations of pediatric participants in order to account for possible background problems with social functioning and negative experiences in real life.We support the view that future studies examining the impact of screen media use in the pediatric population should account for multiple contextual, individual and environmental factors exceeding a one-directional quantity—focused measurements. Scholars may employ new objective tools such as the Comprehensive Assessment of Family Media Exposure (CAFE) to incorporate cultural and socioeconomic factors, as well as assess content and duration of media use [125]. Ideally, more research of experimental methodology would enable scholars to draw causative conclusions.Digital Health Interventions encompass a variety of tools designed to help maintain or regain health. This includes specific applications, video games and exergames targeting symptoms of chronic diseases, neurodevelopmental disorders and mental health problems. The body of research is emerging and steadily growing, yielding preliminary, promising results. However, these applications should present evidence from objective, experimental trials before claiming their positive influence on health interventions or compliance.

## Data Availability

No new data were created or analyzed in this study. Data sharing is not applicable to this article.

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
