# Peer review of "Raising the Child—Do Screen Media Help or Hinder? The Quality over Quantity Hypothesis"

_ijerph, 2022, doi:10.3390/ijerph19169880_

Round 1
Reviewer 1 Report
This is a timely article that raises important considerations of screen use or technology exposure with children. I have a few suggestions as below:
1. The title suggests that it is a debate where both positive and negative effects of screen exposure should be considered with an objective lens. However, the first big half of the article, particularly the fist section, has predominantly focused on the negative impacts of screen use. I suggest buffering each sections slightly with more neutral or positive languages or evidence.
2. Video calls or zoom, which is very common in younger children as well, have not been discussed much. Here are a few suggestions of recent publications for the authors to consider:
-Strouse, G. A., McClure, E., Myers, L. J., Zosh, J. M., Troseth, G. L., Blanchfield, O., ... & Barr, R. (2021). Zooming through development: Using video chat to support family connections. Human Behavior and Emerging Technologies, 3(4), 552-571.
-Sundqvist, A., Koch, F. S., Birberg Thornberg, U., Barr, R., & Heimann, M. (2021). Growing up in a digital world–digital media and the association with the child’s language development at two years of age. Frontiers in psychology, 12, 569920.
-Stuckelman, Z., Strouse, G., Myers, L., Barr, R., Zosh, J. M., McClure, E., ... & Troseth, G. (2022). The Role of Technology Comfort and Access in Grandparent-Grandchild Video-Chat Frequency. Computers in Human Behavior.
3. Lastly, I think the paper should propose explicit, concrete future directions and implications. See these papers with suggestions of considering historical factors while evaluating the effect of screen exposure, as different generation, and people in different areas around the world may very much perceive screen use very differently, depending on various historical as well as environmental and cultural factors - this point should be elaborated and emphasised.
-Nielsen, M., Fong, F. T., & Whiten, A. (2021). Social learning from media: The need for a culturally diachronic developmental psychology. Advances in Child Development and Behavior, 61, 317-334.
This following paper established how children these days may view digital screens as a new source of information conveying important information related to their cultural norms, where they might follow screen information more socially than live information or than information from a platform they are less familiar with. This supports this paper's hypothesis that quality is more important.
-Fong, F. T., Imuta, K., Redshaw, J., & Nielsen, M. (2021). The digital social partner: Preschool children display stronger imitative tendency in screen-based than live learning. Human Behavior and Emerging Technologies, 3(4), 585-594.
This following paper proposes a comprehensive way of measuring screen exposure, considering contextual information rather than quantity of exposure. Again, it supports the authors' hypothesis.
-Barr, R., Kirkorian, H., Radesky, J., Coyne, S., Nichols, D., Blanchfield, O., ... & Fitzpatrick, C. (2020). Beyond screen time: a synergistic approach to a more comprehensive assessment of family media exposure during early childhood. Frontiers in Psychology, 11, 1283.
These are only a few suggestions of recent publications which support the authors' hypothesis. The authors should feel free to include more papers. I think the conclusion paragraph (last section) needs to be strengthened with more contents to enhance the hypothesis proposed in this paper.
Reviewer 2 Report
I'd like to start my feedback on this article by stating that I am a scholar in media and cultural studies and so may not be completely aligned with the disciplinary norms that drive this article.
This is a review article that aims to synthesize the findings of a number of previous studies regarding the impact of screen media consumption on the development and 'health' (broadly defined) of children. It is comprehensively researched and very well organized in terms of structure. Also, since this is a review piece, it is lacking in the presentation of new/original data or findings to the academic community. Instead, the article aims to consolidate existing knowledge arising from different disciplines. It is successful in achieving these aims.
However, previous review articles that I have encountered often not only summarize existing positions within a (sub-)field, but also use this as a way of suggesting new areas and trajectories for future research. Although there are some gestures made to where 'more research is needed' in particular sectors, this article is currently quite lacking in making those assertions. This is a weakness of this article as failure to provide the authors own suggestions for future research means that the paper does not really take a position on any of the issues that are raised. I'd strongly suggest that the authors address this issue in a future draft.
There is also quite a lot of work that could be done on conceptual definitions threaded through the document. The key terms of 'health' and 'child' are vaguely defined throughout and, especially with the latter term, lean towards essentializing the topic. For example, 'health' is used interchangeably here to refer to both physical and mental health conditions when these have different trajectories, institutional definitions, and so on. By fusing all of these together, the nuances of how different forms of 'health' are meaningful becomes lost. Similarly, the term 'child' is moved freely across different national contexts, which imposes the image of a universal and essentialized 'child' who remains the same across multiple social, cultural, and national contexts. This is clearly not the case, and a future draft of this paper should pay greater attention to how this term is being used across different contexts.
Finally, a strong proof-read of the article to eliminate errors in phrasing, grammar, and typos is needed.
Round 2
Reviewer 1 Report
Thank you authors for taking careful considerations of my suggestions. I think the authors have adequately addressed my comments and have nothing further to add. The manuscript is much improved.
Author Response
Dear Sir or Madam,
Thank you for valuing our work.
We appreciate the time and effort that you dedicated to providing us with the comments on the draft. We believe that, by following your advise, we improved our paper.
Kind regards,
Authors
Reviewer 2 Report
Thank you to the authors for the time and dedication that they have dedicated towards tackling the feedback provided. The changes made to this drfat have vastly improved the focus and balance of the arguments made, and the signposting of future directions for research add an extra dimension to the discussion here.
There are only a couple of very minor changes that I'd invite the authors to look into before accepting the article. These are:
Lines 38-41 - the current phrasing of this statement makes it unclear as to who the benefits mentioned belong (e.g., are these to children or to adults?).
Lines 374-5 - I'd query the point made about the suggested 'bias' of 'self-reported data'. Is this a statement about the apparent 'objectivity' of quantitative data in comparison with qualitative equivalents? If so, this is an inaccurate assertion to make, as qualitative data can be just as insightful and 'objective' as quantitative equivalents.
Finally, I'd suggest one final pass over the overall writing style for this piece of work. Although the phrasing is much clearer here, and whilst I'm highly sensitive to English not being the authors' first language, there are still quite a few slippages in use of grammar throughout the document.
As someone who teaches a first-year undergraduate course about media audiences, and which contains a week on issues linked to 'media effects' and influence, I'd strongly consider including this article on the reading list.
Author Response
Dear Reviewer,
Thank you for the appreciation of our work.
We are grateful for the effort you have dedicated to our paper and the comments. We believe the text, by following your suggestions, has been improved.
We have adressed the points that you accentuated in the last review:
1. Lines 38-41 - the current phrasing of this statement makes it unclear as to who the benefits mentioned belong (e.g., are these to children or to adults?).
Unfortunately, we failed to find the mentioned unclear statement concerning "benefits" under the noted lines. However we have checked the whole draft in search for relevant sentences and corrected two sentences in lines 162 and 182.
2. Lines 374-5 - I'd query the point made about the suggested 'bias' of 'self-reported data'. Is this a statement about the apparent 'objectivity' of quantitative data in comparison with qualitative equivalents? If so, this is an inaccurate assertion to make, as qualitative data can be just as insightful and 'objective' as quantitative equivalents.
Thank you for pointing out the ambiguity concerning the mentioned bias. We agree that the nature of gathered data - whether quantitative or qualitative - is irrelevant. We have specified the kind of bias we meant at this point of the manuscript (a social desirability bias).
We would also like to thank you for the understanding of our linguistic difficulties. A native speaker checked the text and corrected it. We hope that the level of linguistic and grammatical accuracy is satisfactory now.
Finally, we do appreciate you kind remark regarding the consideration of utilization of our paper for teaching purposes.
Best regards,
Authors